# Analyzing the Relationship between Participation in Sports and Harmful Alcohol Drinking in Early Adolescence: Two-Year Prospective Analysis

**DOI:** 10.3390/children10061065

**Published:** 2023-06-15

**Authors:** Natasa Zenic, Ivan Kvesic, Mate Corluka, Tatjana Trivic, Patrik Drid, Jose M. Saavedra, Nikola Foretic, Toni Modric, Barbara Gilic

**Affiliations:** 1Faculty of Kinesiology, University of Split, 21000 Split, Croatia; 2Faculty of Science and Education, University of Mostar, 88000 Mostar, Bosnia and Herzegovina; 3Faculty of Sport and Physical Education, University of Novi Sad, 21000 Novi Sad, Serbia; 4Physical Activity, Physical Education, Sport and Health Research Centre, Sports Science Department, School of Social Sciences, Reykjavik University, IS-101 Reykjavik, Iceland

**Keywords:** non-communicable diseases, harmful drinking, AUDIT, puberty, physical exercising

## Abstract

Alcohol drinking is an important health-related problem and one of the major risk factors for a wide array of non-communicable diseases, while there is a lack of studies investigating environment-specific associations between sports participation and alcohol drinking in adolescence. This study prospectively investigated the relationship between sports factors (i.e., participation in sports and competitive achievement), with the prevalence of harmful alcohol drinking (HD), and HD initiation in 14-to-16 years old adolescents from Bosnia and Herzegovina (*n* = 641, 337 females, 43% living in rural community). Participants were tested over 4-time points divided by approximately 6 months, from the beginning of high school to the end of the second grade. Variables included gender, factors related to sport participation, a community of residence (urban or rural), and outcome: alcohol consumption was assessed by the AUDIT questionnaire. Results evidenced that the prevalence of HD increased over the study period from 6 to 19%, with no significant differences between urban and rural youth. Logistic regression for HD as criterion evidenced adolescents who participated in sports and then quit as being at particular risk for drinking alcohol at the study baseline. Sports factors were not correlated with HD initiation in the period between 14 and 16 years of age. It seems that the problem of alcohol drinking should be preventively targeted in all youth, irrespective of living environment. Although sports participation was not evidenced as being a factor of influence on HD initiation, results highlight the necessity of developing targeted preventive campaigns against alcohol drinking for adolescents who quit sports.

## 1. Introduction

Adolescents (i.e., individuals aged 10–19 years) are in a sensitive life phase that includes psychological, physical, and social changes [1]. During this life phase, adolescents shape their lifestyle habits that determine their adulthood choices and health status [2]. Precisely, adolescents are involved in numerous health-threatening behaviors, such as consuming psychoactive substances, including alcohol. Alcohol is one of the major risk factors for a wide array of non-communicable diseases (e.g., digestive diseases, cancer, and cardiovascular-illness) [3]. What is alarming is that it has been evidenced that alcohol drinking in adolescence increases risks for psychopathology in adulthood, including alcohol dependence and different cognitive impairments [4,5]. Thus, it is worrying that adolescents are reported to be highly engaged in harmful alcohol drinking [6]. Drinking alcohol is one of the most prevalent health-risk behaviors in Bosnia and Herzegovina (B&H), and B&H is one of the European countries with the highest rate of alcohol drinking [7]. Namely, 27% of girls and 41% of boys aged 17–19 years have been reported as harmful alcohol drinkers [8]. Thus, adolescents from B&H are especially vulnerable when it comes to drinking alcohol and deteriorating their future health.

Being recognized as an important public-health issue, the prevalence of alcohol consumption and the factors of influence on alcohol consumption are frequently studied, particularly in vulnerable groups, such as adolescents [9,10]. Among other factors, sports participation has repeatedly been correlated with alcohol drinking [11,12]. Indeed, while sport could be recognized as protective against alcohol drinking (i.e., sport is considered as activity which promotes pro-social behavior), there are opposed theories of sport as factor of increased risk for alcohol consumption [13]. In a review study by Kwan, 82% of the included studies reported a positive relationship between sport participation and alcohol drinking in adolescents worldwide [11]. In other words, adolescents who participate in sports activities are found to be more prone to drinking alcohol [11]. It was explained by the fact that drinking alcohol is a social activity, and sport is one of the leading socializing agents [13]. Additionally, gatherings in bars and nightclubs after sports activities and events, media alcohol promotions, and the influence of active peers are probably the most influential factors for drinking alcohol in the sports population, which puts even adolescents involved in sports in higher risk of alcohol drinking [14,15,16].

These findings were confirmed in adolescents living in southeastern Europe [17,18]. However, the beforementioned studies investigated older adolescents, while there is a lack of studies investigating adolescents under the age of 16. Indeed, sports factors were associated with alcohol drinking in adolescents aged 16 years, while such associations were not recorded at the follow-up when adolescents had 18 years [19]. This indicates that adolescent athletes started to drink alcohol even at an earlier age (before 16 years of age). Therefore, the incidence of drinking alcohol at earlier age must be investigated to precisely determine whether younger adolescents/children are susceptible to drinking alcohol so the public health authorities might act to prevent such health-risk behavior on time. The main reason athletes start drinking at an earlier age could be the social and cultural context of drinking alcohol. Briefly, adolescents that participate in sports spend more time out of home, socialize with their peers and older athletes, and are frequently exposed to social situations that include drinking alcohol [13]. Additionally, adolescents living in different environments (e.g., urban and rural) have different opportunities and customs for drinking alcohol, which means that environment also plays a role in the incidence of alcohol drinking among adolescents. Briefly, sport participation and alcohol use were associated, but with a stronger association among urban- compared to rural-living adolescents [20].

From the previous brief literature overview, it is evident that studies already investigated sport as a factor of influence on alcohol drinking in older adolescents from southeastern Europe, including B&H [17,19]. On the other hand, there is a limited number of studies investigating the influence of sports participation on alcohol consumption in younger adolescents. This is particularly important if we consider that previous studies done with 16-to-19 years old adolescents clearly highlighted the necessity of studying factors associated with harmful alcohol drinking in earlier age, emphasizing the possible association which may exist between sport participation and occurrence of alcohol drinking [8,17,19]. Therefore, this research aimed to examine whether there are associations between sports participation and alcohol drinking in 14- to 16-year-old adolescents from B&H. The main hypothesis of this study was that there would exist a positive association between sports factors and alcohol consumption, with a higher likelihood of alcohol drinking in adolescents who participate in sports.

## 2. Materials and Methods

### 2.1. Study Design and Participants

Participants were 641 adolescents from Bosnia and Herzegovina (337 females) of whom 43% lived in a rural community, while 57% lived in an urban community during the study course. All resided in Herzegovina Neretva, Herzeg Bosnian County/Canton 10, and Western Herzegovina in Bosnia and Herzegovina. This investigation was part of a larger investigation, and we used a multi-stage cluster sampling method, including (i) random selection of one-quarter of high schools in the observed Cantons and (ii) random selection of one-third of the first-year classes. With a population of approximately 2600 who were high-school first graders in the previous academic year, a previously reported prevalence of harmful drinking for somewhat older adolescents in the same region (22%), a Type I/II error rate of 0.05, and a statistical power of 80%, the necessary sample size was 302 participants [17]. Participants were tested over the following four waves. In each wave, all variables were assessed. It allowed us to evidence the participants who initiated HD in the period between first and fourth measurements (between the age of 14 and 16 years). The baseline testing was carried out at the beginning of the first year of high school (14 years old on average); follow-up 1 was carried out at the end of the first year of high school (6–7 months later); follow-up 2 was carried out at the beginning of the second year of high school when participants were 15 years old on average, and the fourth wave (follow-up 3) was carried out six months later, at the end of the second year of high school (Figure 1).

One week prior to the first testing wave, investigators explained the study background, idea, and protocol to the children’s parents at regular school meetings and obtained parental written consent for study participation. The study was anonymous, and no personal data were asked, but to follow responses over testing waves, participants were asked to use a self-selected confidential code. The investigation was approved by the Ethical Board of Faculty of Kinesiology, University of Split.

In total, the 678 participants were involved during the baseline testing; however, final analysis included only participants that were tested in all four testing waves and who did not change place of residence during the course of the study. Participants were tested while they were in school classes occurring at the beginning of the school day (i.e., their first lesson). Each participant was supplied with the questionnaire and one envelope. After completing the questionnaire, each participant placed the questionnaire in an envelope and then placed the envelope in a closed box that was opened the next day by an investigator who was not present at testing. Prior to testing, they were informed that they can refuse to participate and leave some questions or the entire questionnaire unanswered.

### 2.2. Variables

The variables in this study included age (in years), sex (male, female), sport participation, community of residence (participants reported place of residence which was later classified according to national statistics into urban or rural), and outcome: alcohol consumption and potentially problematic drinking as assessed by the Alcohol Use Disorders Identification Test (AUDIT).

AUDIT is a questionnaire repeatedly found to be a reliable and valid measuring tool which asks the subject about 10 items (e.g., How often do you have a drink containing alcohol? During the past year, how often have you had a feeling of guilt or remorse after drinking?, etc.) [21]. Scores range from 0 to 4 for a hypothetical minimum (0) to maximum (40) range (please see Appendix A for AUDIT questionnaire used in the study). Apart from total AUDIT score, for the purpose of statistical analyses in this study we observed HD and the initiation of HD during the study course. AUDIT was developed for screening problems related to alcohol (e.g., harmful alcohol consumption) in the primary care setting. The purpose of the questionnaire was to enable early and effective interventions for dealing with problematic alcohol use. Indeed, any drinking by early adolescents and binge drinking by middle adolescents has to be screened, and preventive interventions should be provided [22]. Thus, the AUDIT was used to identify mostly the prevalence of drinking alcohol (e.g., question “How often do you have six or more drinks on one occasion?”) and what alcohol does to the person (e.g., question “During the past year, how often have you failed to do what was normally expected of you because of drinking?”) and not types of alcoholic beverages that adolescents drink in order to identify individuals who have problems with drinking [23]. Specifically, the results were divided into “harmful drinking” (HD; scores of 8 or above) and “non-harmful drinking” (NHD; scores below 8) [24]. HD relates to alcohol problem use, depicting the amount and prevalence of drinking alcohol that has harmful effects on health [25]. Even though it was advised to use even a lower cut-off point (i.e., scores below 5 were considered as HD) [25], we used the higher cut-off point to allow meaningful comparison with results of the previous studies which examined somewhat older adolescents in the country [17,19].

Sport participation was assessed with scale which was repeatedly used in studies in the region and included the following questions: (i) participation in sports with answers never involved-quit-currently involved, (ii) personal best competitive result/achievement answered of 3-point scale (never involved/competed-local competitions-national/international competitions), (iii) years of sports involvement on a 4-point scale from never involved to involved for more than 5 years, and (iv) number of training sessions per week on 3-point scale from “one-session per week” to “almost every day” [26].

### 2.3. Statistics

All variables were checked for normality by Kolmogorov–Smirnov test. For normally distributed variables, means and standard deviations were reported, while frequencies (counts) and percentages were presented for nonparametric variables.

Analysis of variance for repeated measurements was calculated to evidence the differences among measurements in AUDIT raw score. The Chi-square test was applied to evidence the differences in HD between urban and rural youth.

Associations between studied variables and outcomes (HD and initiation of HD) were assessed by logistic regression analysis. Specifically, logistic regressions were calculated between predictors (sports factors) and outcome (HD) in each wave. Additionally, another logistic regression was calculated between sports factors obtained at the study baseline and HD initiation over the study course (for those participants who did not report HD at the study baseline). Logistic regressions were calculated with urban/rural environment as a covariate, with the intention to calculate additional environment-stratified analyses if urban/rural living would be evidenced as a significant confounding factor (please see Results for more details). Odds Ratio and 95% Confidence Interval were calculated, while the Hosmer Lemeshow test was applied for checking the model fit.

## 3. Results

Results of the AUDIT scores with differences among testing points are presented in Table 1. In brief, alcohol consumption as evidenced by AUDIT scale significantly increased for total sample and when observed separately for each living environment. No evident differences in trends of changes were observed in urban vs. rural youth.

Data on changes in HD over the study period are presented in Figure 2. In brief, the prevalence of HD increased over the study period, from 6% adolescents who reported HD at study baseline to 19% of adolescents who reported HD at fourth testing wave (end of 2nd year of high school). However, no statistical differences in the prevalence of HD were evidenced between youth living in urban and those living in rural communities (Chi-square: 1.34–4.11; *p* > 0.05).

Table 2 presents the results of the logistic regression calculation for the HD criterion at the study baseline. Results indicate a significant association between quitting sports and HD with a higher likelihood of HD in those adolescents who participated and then quit sports. Meanwhile, current participation in sports was found to be a significant protective factor against HD, but this association did not reach statistical significance when regression included urban/rural living environment as a covariate.

No significant association between sports factors and HD prevalence was evidenced at the first follow-up (at the end of the first grade of high school) (Table 3).

Sport factors were not correlated with HD at second follow-up measurement (Table 4).

Logistic regression did not evidence a significant association between sport factors and HD at the fourth testing wave, at the end of the second grade of high school (Table 5).

When logistic regression was calculated for HD initiation during the study course, sports factors were not significantly associated with the criterion variable (Table 6).

## 4. Discussion

This study aimed to investigate whether there are associations between sports participation and alcohol drinking (e.g., HD and HD-initiation) in younger adolescents from B&H. Results revealed several important findings. First, there was a high prevalence of HD in young adolescents. Second, results evidenced a higher occurrence of HD at the start of high school among those adolescents who quit sports, but a lower occurrence of HD at the start of high school was evidenced for adolescents being involved in sports. Third, sports factors obtained at the study baseline were not correlated to HD initiation over the study course. Therefore, we cannot accept our initial study hypothesis.

### 4.1. Prevalence of Alcohol Consumption in Early Adolescence

The first important finding of this research is that alcohol consumption significantly increases from the beginning of high school to the end of the second grade (i.e., from 14 to 16 years of age). Moreover, HD also increases significantly, from 6% at baseline, to almost 20% of adolescents aged 16 years who reported HD. Therefore, the idea of previous studies that alcohol drinking begins at a very early age (i.e., during early adolescence) was confirmed [19].

Studies conducted on the territory of B&H and border countries reported a high prevalence of HD in adolescents aged 16–19 years. Among adolescents from B&H aged 17–18, 41% of boys and 19% of girls were considered harmful alcohol drinkers [8]. Similarly, 22% and 29% of Croatian adolescents aged 16 and 18 were reported as harmful alcohol drinkers, respectively [17]. Additionally, 43% of boys and 39% of girls from Kosovo were heavy drinkers [18]. Thus, such an alarming trend of a rapid increase in the prevalence of HD over the adolescent years is proven in this study. What is more worrying, regarding the high prevalence of alcohol drinking, it is evident that the majority of adolescents initiate with HD at an earlier age (14–16 years).

Supportively, a Brazilian study noted that the average age of drinking onset was 10.8 years [27]. Moreover, an Australian study that followed adolescents over seven years, from 12 to 19 years of age, reported a rapid increase in drinking from 12 years of age which was related to alcohol abuse in later adolescence to 19 years of age [28]. HD was prevalent among 46.8% of Norwegian and 38.9% of young Australian adults aged 19–25 years, which was influenced by drinking behaviors in early adolescence at ages 14–16 years [29]. Thus, our finding that alcohol consumption increases rapidly from early adolescence is logical. What is more worrying, drinking in early adolescence influences high prevalence of drinking in late adolescence and adulthood [29].

Due to different settings, life opportunities, and socioeconomic status, one could expect differences in lifestyle factors such as sports participation and alcohol consumption among individuals living in different environments (i.e., urban and rural living environments). Indeed, a study by Hoffmann [20] recorded that sports participation was associated with alcohol use, with a stronger relationship among youth living in higher socioeconomic neighborhoods (i.e., urban community). This could be explained by the fact that adolescents who live in urban communities have higher accessibility to sports facilities and organized activities [30,31]. Additionally, urban adolescents probably have better accessibility to bars and clubs that provide alcoholic drinks and a better financial status, which allows them to purchase alcoholic beverages. Hence, we expected that the living environment can be a factor that influences relationships between sport factors and HD as well. However, we did not find evidence of such considerations. Even more, it seems that there is no difference in alcohol consumption between adolescents living in the urban or rural environments. Supportively to our findings, a study on adolescents and young adults aged 14–25 did not find urban–rural differences in drinking alcohol [32]. In our study, the main reason for a similar prevalence of alcohol drinking in urban and rural youth is probably related to the fact that alcohol prevalence in B&H is generally high, which logically resulted in a high prevalence of alcohol drinking both among rural and urban adolescents, irrespective of the living environment. The similarity in prevalence of HD probably resulted even in similar relationship between sport factors and HD in urban- and rural-youth (please see following discussion).

### 4.2. Sport Participation and Alcohol Consumption

Given that participation in sports is known to be related to positive developmental outcomes, such as self-esteem, academic achievement, and physical health, sports are considered to be a mechanism of encouraging pro-social behavior [33]. Consequently, active sports participation is often considered a protective factor against adolescent substance misuse, and the relationship between sport participation and substance misuse, including alcohol consumption, has been investigated [26]. However, while some studies reported higher levels of alcohol consumption in adolescents who were engaged in some form of sport compared to those who were not involved in sports activities, other studies highlighted sports participation as a protective factor against alcohol use [14,34]. These inconsistencies suggest that participating in sports does not always increase an individual’s vulnerability to alcohol use and that sports are protective against alcohol use in some cases, while in others sport should be observed as a risk factor for substance misuse [35].

Our findings at least partially support such considerations, indicating sport participation as a factor specifically related to alcohol consumption in adolescents. We evidenced a higher occurrence of HD at the start of high school (i.e., study baseline) among those adolescents who participated in but eventually quit sports. It seems that quitting sports before high school may be a risk factor for adolescents to be involved in HD at the start of high school. Supportively, similar results were evidenced in a previous study that investigated adolescents at the age of 16 and reported that quitting sports was a risk factor for HD in late adolescence [17]. However, it is noteworthy that this trend was not observed in any of the three follow-ups in the current study, although HD has increased in each follow-up (please see previous Discussion). Most probably, other socio-demographic, familial, or school factors possibly contributed to increasing HD in this period of life (e.g., early adolescence), consequently decreasing the effect of quitting sports as a risk factor for HD.

Our results point to another important relationship. Briefly, lower occurrence of HD at the start of the high school was evidenced for adolescents being involved in sports, indicating that sport participation at the beginning of high school may be protective factor against HD. Such results are in contrast to the study that reported a higher occurrence of HD among 16 years old adolescents who were more involved in sports (i.e., had a higher number of training sessions per week) [19]. This emphasizes that the influence of sport participation on alcohol consumption changes over the years. Even in our study, where adolescents were tested in four testing waves, the associations between sports factors and HD prevalence changed considerably comparing study baseline and follow-ups (i.e., sports participation was not found to be a protective factor against HD in all three follow-ups).

All previously said can be additionally confirmed by the results of a prospective analysis. In brief, sports factors obtained at the study baseline (at the beginning of high school education) were not correlated to HD initiation over the study course. It must be mentioned that in these analyses, we did not include those adolescents who reported HD at the study baseline. Therefore, it is clear that factors related to HD-initiation after the age of 14 years should be investigated outside of the sport context. However, irrespective of the non-significant association between sports factors and HD initiation, the fact that sport was not evidenced as protective against HD deserves attention. In short, despite the common perception that sports participation develops pro-social behavior, it is simultaneously associated with certain risks of higher alcohol consumption in youth. With this regard, a specific socio-cultural environment of sport deserves attention.

In a sport environment, alcohol consumption is common, and in some cases, it is set as a socio-cultural imperative [36]. Young athletes often find themselves in situations where alcohol is consumed, and it is difficult for them to remain spared from the influence of the environment, and “in their defence,” they start consuming alcohol themselves [37]. Some studies present the common causes of such phenomena. First, it is emphasized that the consumption of some types of alcoholic beverages (primarily beer) is socially accepted and even encouraged in sports environments. Likewise, it is not uncommon for beer consumption to be perceived as a way to improve recovery after strenuous sports training or competition [38]. Second, alcohol is the most widespread sedative in the world, and its consumption, to a certain extent, contributes to mental and physical relaxation. Therefore, it is not surprising that athletes often consume it after training or stressful competitions. The third possible explanation is related to the specifics of the region where the sample was drawn. In brief, Bosnia and Herzegovina is actually a Mediterranean country where alcohol consumption is common and widespread [26]. One can argue about the Islamic religion (a significant part of the population and Islamic Muslims) and the known boundaries of the Islamic religion against alcohol. However, studies conducted so far confirmed that religious affiliation in this country is not related to lower alcohol consumption, at least not in youth [26,39].

To sum up, this study recorded an alarming incidence of students that drink alcohol starting from an early age. Public health authorities should be informed on these findings in order to make preventive interventions to stop adolescents from early start of drinking alcohol, which is particularly important for adolescents who quit sports who were found to be in risk for early onset of HD. Therefore, apart from public health authorities our results should be spread among sport clubs and schools, with the message that youth should continue practicing sports in order to stay away from hazardous behaviors such as alcohol drinking. However, to do so, all stakeholders, including health and educational authorities, should make sports more accessible to everyone and enable participation for individuals that are not willing or able to partake in sports competitions but would like to do sports recreationally.

### 4.3. Limitations and Strengths

The main limitation of the study is related to the fact that variables were self-reported. Therefore, it is possible that participants leaned toward socially desirable answers, which could be particularly possible for alcohol consumption. However, when compared to other studies conducted in the region that used the same measurement tools and studied somewhat older adolescents, it seems that data reported here are plausible [8,17]. Additionally, our results highlighted very complex relationships between sport participation and alcohol drinking, while we used a relatively limited number of variables examining sport factors, mainly because of the prospective repeated measurement design. Therefore, in future studies more detailed analysis of the sport factors in relation to substance misuse behaviors is needed, and special attention should be placed on the type of sport adolescents are involved at (i.e., team-sports, individual-sports, martial-arts, or aesthetic sports). In this study, the AUDIT score of 8 was considered as a cut-off point for HD, although a cut-off point of 5 was suggested for younger adolescents [25]. However, this was conducted intentionally to allow meaningful comparison with previous studies done with older adolescents from the country [17,19]. Finally, in this study we did not specifically observe eventual influence of gender on studied relationship (i.e., analyses were not gender stratified), but this was due to the fact that we observed participants throughout several time points. Therefore, in further studies specific gender-stratified analyses on a problem are warranted.

Despite limitations, the study has several strengths. Most importantly, this is one of the rare prospective studies examining the problem of the relationship between sport-factors and alcohol drinking, while systematically observing potential problems of the urban/rural environment among younger adolescents. Additionally, this is probably the first investigation that examined the problem in southeastern Europe, a region with a high prevalence of alcohol drinking among adolescents. Therefore, we believe that our findings will increase knowledge in the field and initiate further analyses.

## 5. Conclusions

This study evidenced certain associations between sports factors and alcohol drinking in adolescence, once again confirming quitting sports as a factor of increased risk for HD in adolescence, but this time emphasizing this problem in younger adolescence. Evidently, children who participated in sport and then quit are at higher risk for HD. Therefore, it is clear that sport- and public-health authorities should be specifically warned about it to develop accurate and targeted preventive campaigns for people of young age.

It seems that the problem of alcohol drinking in B&H should be similarly targeted in all youth irrespective of living environment. The similar prevalence of HD and trajectories of changes in alcohol drinking between communities additionally confirm such a conclusion.

Future studies should focus on other factors potentially related to alcohol drinking in adolescents from the country and region. This is particularly important if we take into account that our results showed an alarming increase in alcohol consumption in the period between 14 and 16 years of age.

## Figures and Tables

**Figure 1 children-10-01065-f001:**
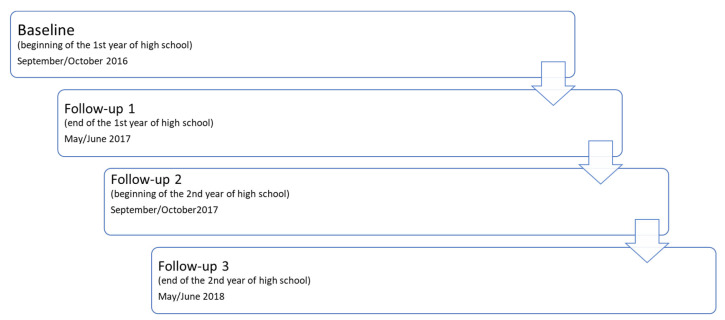
Study design.

**Figure 2 children-10-01065-f002:**
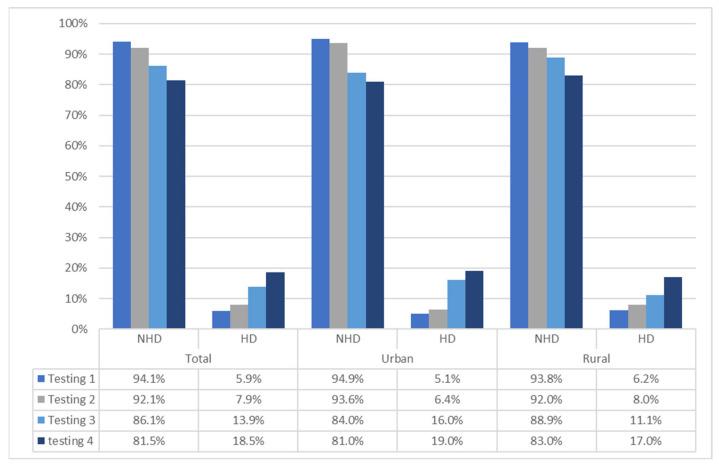
Prevalence of harmful drinking (HD) and nonharmful drinking (NHD) over the study period in the total sample, urban and rural youth.

**Table 1 children-10-01065-t001:** AUDIT scores and differences among testing waves calculated by analysis of variance (ANOVA).

	1st Testing		2nd Testing		3rd Testing		4th Testing		ANOVA	
	Mean	SD	Mean	SD	Mean	SD	Mean	SD	F Test	*p*
Total	1.50	0.30	1.82	0.32	2.74	0.36	3.51	0.40	79.97	0.001
Urban	1.51	0.50	1.80	0.51	2.71	0.57	3.53	0.61	28.11	0.001
Rural	1.48	0.60	1.83	0.61	2.76	0.62	3.49	0.60	25.41	0.001

**Table 2 children-10-01065-t002:** Logistic regression for criterion harmful alcohol drinking at baseline (gender and urban/rural living environment were included as covariates in Model 1).

	Model 0	Model 1
Predictors	**OR (95%CI)**	**OR (95%CI)**
Participation in sport		
Yes, I am participating	2.46 (1.02–5.96)	2.08 (0.84–5.13)
Quit	2.52 (1.14–5.55)	2.43 (1.09–5.44)
Never	REF	REF
Sports involvement (years) ^ordinal^	1.1 (0.81–1.49)	0.98 (0.71–1.34)
Competitive achievement/result ^ordinal^	1.14 (0.79–1.64)	1.1 (0.71–1.49)
Number of training sessions per week ^ordinal^	1.37 (0.86–2.17)	1.28 (0.8–2.04)

**Table 3 children-10-01065-t003:** Logistic regression for criterion harmful alcohol drinking at first follow-up (gender and urban/rural living environment were included as covariates in Model 1).

	Model 0	Model 1
Predictors	**OR (95%CI)**	**OR (95%CI)**
Participation in sport		
Yes, I am participating	1.56 (0.73–3.35)	1.26 (0.58–2.74)
Quit	1.42 (0.75–2.72)	1.27 (0.66–2.45)
Never	REF	REF
Sports involvement (years) ^ordinal^	1.28 (0.98–1.69)	1.14 (0.86–1.49)
Competitive achievement/result ^ordinal^	1.17 (0.85–1.59)	1.06 (0.77–1.47)
Number of training sessions per week ^ordinal^	1.67 (1.01–2.61)	1.46 (0.92–2.32)

**Table 4 children-10-01065-t004:** Logistic regression for criterion harmful alcohol drinking at second follow-up (gender and urban/rural living environment were included as covariates in Model 1).

	Model 0	Model 1
Predictors	**OR (95%CI)**	**OR (95%CI)**
Participation in sport		
Yes, I am participating	1.5 (0.78–2.88)	1.25 (0.64–2.44)
Quit	1.36 (0.83–2.23)	1.21 (0.74–2.00)
Never	REF	REF
Sport involvement (years) ^ordinal^	1.09 (0.89–1.33)	0.97 (0.79–1.2)
Competitive achievement/result ^ordinal^	1.16 (0.91–1.48)	1.05 (0.82–1.37)
Number of training sessions per week ^ordinal^	1.29 (0.86–1.95)	1.21 (0.79–1.85)

**Table 5 children-10-01065-t005:** Logistic regression for criterion harmful alcohol drinking at third follow-up (gender and urban/rural living environment were included as covariates in Model 1).

	Model 0	Model 1
Predictors	**OR (95%CI)**	**OR (95%CI)**
Participation in sport		
Yes, I am participating	1.12 (0.62–2.02)	1.09 (0.6–1.99)
Quit	1.52 (0.99–2.27)	1.48 (0.99–2.22)
Never	REF	REF
Sports involvement (years) ^ordinal^	0.91 (0.77–1.07)	0.88 (0.75–1.04)
Competitive achievement/result ^ordinal^	0.88 (0.71–1.1)	0.85 (0.68–1.07)
Number of training sessions per week ^ordinal^	1.06 (0.77–1.46)	1.08 (0.78–1.5)

**Table 6 children-10-01065-t006:** Logistic regression for criterion “harmful alcohol drinking initiation” during the study course (gender and urban/rural living environment were included as covariates in Model 1).

	Model 0	Model 1
Predictors	**OR (95%CI)**	**OR (95%CI)**
Participation in sport		
Yes, I am participating	1.12 (0.62–2.02)	1.09 (0.6–1.99)
Quit	1.52 (0.98–2.27)	1.48 (0.99–2.22)
Never	REF	REF
Sports involvement (years) ^ordinal^	0.91 (0.77–1.07)	0.88 (0.75–1.04)
Competitive achievement/result ^ordinal^	0.88 (0.71–1.1)	0.85 (0.68–1.07)
Number of training sessions per week ^ordinal^	1.06 (0.77–1.46)	1.08 (0.78–1.5)

## Data Availability

Data are available to all interested parties upon reasonable request.

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
