# Peer review of "Analyzing the Relationship between Participation in Sports and Harmful Alcohol Drinking in Early Adolescence: Two-Year Prospective Analysis"

_children, 2023, doi:10.3390/children10061065_

Round 1
Reviewer 1 Report
You did some good research, congratulations. It is only necessary to improve a few points.
Greetings.
The information provided between lines 43 and 46 should be supported by authors.
in line 76 they indicate that there is a lack of studies in adolescents under 16 years of age. Here they must add information that highlights the importance of investigating at that age, what could happen if it is not investigated?, what risks does it bring? This information will support what is stated in lines 87 to 89.
on line87 they talk about previous studies, but there is only one reference that supports it, where are all the previous studies? (add them or fix wording).
Among its results is the variable of living in a rural or urban environment, but the introduction does not present information in this regard. They should add what the scientific evidence exposes in this regard.
Line 197. Should it be figure 2?
Reorder the location of tables and figures, so that they are seen after they are named in the text, since line 202 speaks of table 1 but table 3 is seen below. The same for line 213 with table 3.
How could the limitation of self-reported measurement be avoided in the future? (especially alcohol)
I think a paragraph should be added to the discussion explaining the practical utility of the results found, what can be done with those results (what could they be used for, what measures could be taken based on what was found, etc).
Author Response
You did some good research, congratulations. It is only necessary to improve a few points.
Greetings.
RESPONSE: Thank you for recognizing the importance and quality of our work, we appreciate it! Please see below how we improved our manuscript according to your comments.
The information provided between lines 43 and 46 should be supported by authors.
RESPONSE: The reference for the statements in mentioned lines is now added. Please see reference number 3.
in line 76 they indicate that there is a lack of studies in adolescents under 16 years of age. Here they must add information that highlights the importance of investigating at that age, what could happen if it is not investigated?, what risks does it bring? This information will support what is stated in lines 87 to 89.
RESPONSE: Thank you for this valuable suggestion. The sentence that highlights the importance of investigating alcohol drinking before 16 years of age has been added, text now reads: “This indicates that adolescent athletes started to drink alcohol even at an earlier age (before 16 years of age). Therefore, the prevalence and incidence of drinking alcohol at earlier age must be investigated to precisely determine whether younger adolescents/children are susceptible to drinking alcohol so the public health authorities might act to prevent such health-risk behaviour on time.” (please see lines 77-82)
on line87 they talk about previous studies, but there is only one reference that supports it, where are all the previous studies? (add them or fix wording).
RESPONSE: Two more previous studies have been added, please see references number 8 and 14.
Among its results is the variable of living in a rural or urban environment, but the introduction does not present information in this regard. They should add what the scientific evidence exposes in this regard.
RESPONSE: Thank you for noticing this. Sentences regarding environmental effects on alcohol drinking have been added, text now reads: “Also, adolescents living in the different environment (e.g., urban and rural) have different opportunities and customs for drinking alcohol, which means that environment also plays a role in the incidence of alcohol drinking among adolescents. Briefly, sport participation and alcohol use were associated but with a stronger association among urban compared to rural-living adolescents [18].” (please see lines 85-90)
Also, it has been explained in more detail in the Discussion, please see the last paragraph of the section 4.1.
Line 197. Should it be figure 2?
RESPONSE: Yes, thank you for noticing this typing mistake, it is now changed to Figure 2.
Reorder the location of tables and figures, so that they are seen after they are named in the text, since line 202 speaks of table 1 but table 3 is seen below. The same for line 213 with table 3.
RESPONSE: Reordered, thank you.
How could the limitation of self-reported measurement be avoided in the future? (especially alcohol)
RESPONSE: Unfortunately, as far as the authors are aware, no study has used other measure for assessing alcohol consumption such as direct measurement of alcohol (e.g., amount of alcohol in the air that the person breathes out) are not possible to conduct due to relatively short time that alcohol could be detected in the body after consuming it. Thus, we believe that the self-reporting measurement of alcohol consumption could be only more precise, by making the responses anonymous so the participants of the investigation could give more precise and honest answers, as we did in our study.
I think a paragraph should be added to the discussion explaining the practical utility of the results found, what can be done with those results (what could they be used for, what measures could be taken based on what was found, etc).
RESPONSE: The paragraph you suggested to add is now added in the discussion section, the last paragraph. Text reads: “To sum up, this study recorded an alarming incidence of students that drink alcohol starting from an early age. Public health authorities should be informed on these findings in order to make preventive interventions to stop adolescents from early start of drinking alcohol, which is particularly important for adolescents who quit sports who were found to be in risk for early onset of HD. Therefore, apart from public health authorities our results should be spread among sport clubs and schools, with the message that youth should continue practicing sports in order to stay away from hazardous behaviours such as alcohol drinking. However, to do so, all stakeholders, including health and educational authorities should make sports more accessible to everyone and enable participation for individuals that are not willing or able to partake sports competitions but would like to do sports recreationally.” (please see lines 375-380).
Staying at your disposal, and thank you once again.
Reviewer 2 Report
This is a clear and well-written manuscript evaluating the associations between sports participation and alcohol drinking in adolescents from 14 to 16 years old in Bosnia and Herzegovina. The content is original, the statistical analysis seems correct, and the sample size is large. However, an important limitation, later listed by the authors, is that the data were collected through self-reported questionnaires. Moreover, some topics deserve further investigation. Notably, despite the importance that gender has concerning both sports and drinking, it only enters as a covariate in the analyses without any gender comparisons being made. I think an integration from this point of view would be helpful to the study. Another point to be clarified is the importance of the type and amount of alcohol consumed: should wine, beer, and spirits be placed on the same level? And so is a person who drinks light or moderate amounts compared to one who abuses alcohol? Clarify this aspect.
Minor issues:
· Verify the sentence beginning at line 64.
· I suggest reporting AUDIT in the supplementary material.
· Explain what you mean by harmful and non-harmful drinking (line 151).
· You specified that "For normally distributed variables, means and standard deviations were reported”. What does it mean? What statistical parameters did you calculate for the non-normal variables instead? Were there any?
none
Author Response
This is a clear and well-written manuscript evaluating the associations between sports participation and alcohol drinking in adolescents from 14 to 16 years old in Bosnia and Herzegovina. The content is original, the statistical analysis seems correct, and the sample size is large. However, an important limitation, later listed by the authors, is that the data were collected through self-reported questionnaires.
RESPONSE: Thank you for recognizing the importance and quality of our work. Also, thank you for your comments. We tried to follow it and made amendments as suggested.
Moreover, some topics deserve further investigation. Notably, despite the importance that gender has concerning both sports and drinking, it only enters as a covariate in the analyses without any gender comparisons being made. I think an integration from this point of view would be helpful to the study.
RESPONSE: Thank you for your suggestion. Indeed, gender needs to be specifically studied with regard to this topic. However, we were of the opinion that in this study it will make results and analyses to dense, and difficult to organize. Therefore, we will certainly pay attention on it in future studies, and investigate the problem in more details. It is stated as one of the study limitations. Text reads: “Finally, in this study we did not specifically observed eventual influence of gender on studied relationship (i.e. analyses were not gender stratified), but this was due to the fact that we observed participants throughout several time points. Therefore, in further studies specific gender-stratified analyses on a problem are warranted”. (please see Limitations at the end of the Discussion section)
Another point to be clarified is the importance of the type and amount of alcohol consumed: should wine, beer, and spirits be placed on the same level? And so is a person who drinks light or moderate amounts compared to one who abuses alcohol? Clarify this aspect.
RESPONSE: Thank you for this point of view. It relates to the questionnaire we used in this study. Precisely, AUDIT was developed for screening problems related to alcohol (e.g., harmful alcohol consumption) in the primary care setting. The purpose of the questionnaire was to enable early and effective interventions for dealing with problematic alcohol use. Indeed, any drinking by early adolescents and binge drinking by middle adolescents has to be screened and preventive interventions should be provided. Thus, the AUDIT was used to identify mostly the prevalence of drinking alcohol (e.g., question How often do you have six or more drinks on one occasion?”) and what alcohol does to the person (e.g., question “During the past year, how often have you failed to do what was normally expected of you because of drinking?”) and not types of alcoholic beverages that adolescents drink in order to identify individuals who have problems with drinking. This explanation is now added in the Methods section, Variables.
Minor issues:
Verify the sentence beginning at line 64.
RESPONSE: Thank you for noticing, reference is now added. Please see:
Kwan, M.; Bobko, S.; Faulkner, G.; Donnelly, P.; Cairney, J. Sport participation and alcohol and illicit drug use in adolescents and young adults: a systematic review of longitudinal studies. Addict Behav 2014, 39, 497-506, doi:10.1016/j.addbeh.2013.11.006.
I suggest reporting AUDIT in the supplementary material
RESPONSE: AUDIT is now added as supplementary material
Explain what you mean by harmful and non-harmful drinking (line 151).
RESPONSE: The sentence which explains what harmful drinking relates to is now added. Text now reads: “HD relates to alcohol problem use, depicting the amount and prevalence of drinking alcohol that has harmful effects on health [21].” (please see line 121-122)
You specified that "For normally distributed variables, means and standard deviations were reported”. What does it mean? What statistical parameters did you calculate for the non-normal variables instead? Were there any?
RESPONSE: Yes, that is correct. We reported frequencies (counts) and percentages for non-normally distributed variables. It is stated in the Statistics now (please see first sentence of the Statistics subsection)
Thank you once again!